# The transient impact of the African monsoon on Plio-Pleistocene Mediterranean sediments

Bas de Boer[1], Marit Peters[2,3], and Lucas J. Lourens[2]

[1]Earth and Climate Cluster, Faculty of Science, Vrije Universiteit Amsterdam, Amsterdam, the Netherlands
[2]Department of Earth Sciences, Faculty of Geosciences, Utrecht University, Utrecht, the Netherlands
[3]Now at: United Experts cvba, Beringen, Belgium

**Correspondence:** Lucas Lourens (l.j.lourens@uu.nl)

**Abstract.** Over the Plio-Pleistocene interval a strong linkage exists between Northern African climate changes with the supply of dust over the surrounding oceans and continental runoff towards the Mediterranean Sea. Both these signatures in the sedimentary record are determined by orbital cycles influencing on the one hand glacial variability and on the other hand Northern African monsoon intensity. In this paper, we use the intermediate complexity model CLIMBER-2 to simulate African climate during the Plio-Pleistocene between 3.2 and 2.3 million years ago (Ma) and compare our simulations with existing and new climate reconstructions. The CLIMBER-2 model is externally forced with atmospheric $CO_2$ concentrations, ice-sheet topography and orbital variations, all of which strongly influence climate during the Pliocene and Pleistocene. Our simulations indicate that the records of Northern Africa climate oscillate in phase with climatic precession. For the Earth's obliquity cycle, the time lag between the 41,000-year component in insolation forcing and the climatic response increased after inception of Northern Hemisphere (NH) glaciation around 2.8 Ma. To test the outcome of our simulations, we have put emphasis on the comparison between the simulated runoff of grid boxes encompassing the Sahara desert and the Sahel region and the sedimentary records of marine sediment cores ODP Site 659 (Atlantic Ocean) and ODP Site 967 (Mediterranean). In this study we will show for the first time an extended Ti/Al record of Site 967 down to 3.2 Ma. This record strongly correlates with runoff in the Sahara and Sahel regions, whereas correlation with the dust record of Site 659 is moderate and slightly improves after NH ice-sheet inception. We investigated the transient variability of the individual and combined contributions of the Sahel and Sahara regions and found significant transient behaviour, overlapping with inception of NH ice sheets (2.8 Ma) and the Plio-Pleistocene transition (2.6 Ma). Prior to 2.8 Ma, a larger contribution from the Sahara region is required to explain variability of Mediterranean dust input. After this transition, we found that a more equal contribution of the two regions is required, representing an increased influence of Sahel runoff and wet periods.

## 1 Introduction

It is generally accepted that climate is influenced by the orbital parameters precession, obliquity, and eccentricity, after Milankovitch laid a vital foundation still used today. Ever since, astronomical calibration of climatic proxy records and sedimentary cyclicity have been used to reconstruct and understand past climatic variability over the globe (e.g. Hays et al., 1976; Hilgen, 1991; Lisiecki and Raymo, 2005). With use of the astronomical solutions of Laskar et al. (2004), the climatic response

to orbital variations can be determined. Several past climate transitions are strongly linked to orbital changes, such as the inception of NH glaciation (Lisiecki and Raymo, 2007; Bartoli et al., 2011; Bailey et al., 2013; Willeit et al., 2015). Although radiative forcing of orbital variations is too small to force the world into or out of a glacial state with significant ice sheets, they are key to initiate ice-sheet growth and to pace glaciations (e.g. Bintanja and Van de Wal, 2008; Ganopolski and Calov, 2011). If orbital variations can be seen as triggers of glaciation, internal feedback mechanisms causing changes in atmospheric

concentrations of greenhouse gases (GHG) and the enhanced growth of Northern Hemisphere (NH) ice sheets are the main amplifiers of glacial-interglacial climate change. A key driver of intensification of ice-sheet changes is the atmospheric concentrations of $CO_2$ (e.g. Bartoli et al., 2011; van de Wal et al., 2011; Willeit et al., 2019), which over the past 800,000 years (800 kyr) is largely in sync with atmospheric temperature and ice volume (e.g. Lüthi et al., 2008; Stap et al., 2014).

A main transition in the recent geological past is the inception of NH glaciation ($\sim$2.8 Ma) (e.g. Flesche Kleiven et al.,

2002), close to the Plio-Pleistocene transition at $\sim$2.6 Ma. The inception can be related to a drawdown of atmospheric $CO_2$ concentrations (e.g. Willeit et al., 2015; Tan et al., 2018). Although driven largely by the NH, the transition is a global feature, leading to Arctic cooling (Brigham-Grette et al., 2013) and cooling of tropical sea surface temperatures (Herbert et al., 2010). At that time, Earth's climate followed obliquity periodicity with symmetric glacial cycles of $\sim$41 kyr, driven by high latitude climate variability (e.g. Venti et al., 2013). The orbital-induced variability is not limited to the high latitudes, it is also seen

over the entire African continent. North African monsoonal records are linked with runoff and precipitation (e.g. Lourens et al., 2010), which persisted throughout the Pleistocene (Wagner et al., 2019). Changes are seen during the late Pliocene and mid-Pleistocene in vegetation in northeast Africa (Rose et al., 2016), during the Pleistocene in Kenya (Lupien et al., 2018), the West African monsoon (Kuechler et al., 2018) and hydroclimate variability in southeastern Africa over the past 2 million years (Caley et al., 2018).

Another common association with African Plio-Pleistocene climate variability is early hominin evolution in East Africa. Orbital forcing, and its role on climate variability, has long been assumed in many paleoanthropological studies to be a strong influence on early hominins (e.g. deMenocal, 1995; Maslin et al., 2014; Joordens et al., 2019). Specific climate transitions, including the Plio-Pleistocene transition, coincide with the possible emergence or extinction of hominin species (Donges et al., 2011). The earliest hominins appear $\sim$4.5 Ma in Africa, and the first occurrence of the genus Homo appears around 2.8 Ma

(DiMaggio et al., 2015). For example, Joordens et al. (2019) includes the role of spatial distribution and geography in the study of hominin evolution and dispersal. Condition of highly variable climate and strong seasonality during eccentricity maxima would result in isolated refugium for early hominins, that would be conclusive for evolution (Trauth et al., 2007). From these refuges, the evolved hominins would then disperse inland through vegetated corridors during periods of stable climate with low seasonality during eccentricity minima (Joordens et al., 2019). However, the evolution of the hominin species throughout

the Pleistocene is a highly complex process (Mounier and Mirazón Lahr, 2019).

In this paper we focus on the connection between the African monsoon, using continental runoff (linked to precipitation, evaporation, and water storage in the soil, lakes and groundwater) and sediment- and dust-deposition from the African continent between 3.2 to 2.3 Ma, which includes the Plio-Pleistocene transition at 2.6 Ma. Here, we use a model of intermediate complexity; CLIMBER-2 (Petoukhov et al., 2000) to simulate climate variability. Due to its relatively low resolution and low

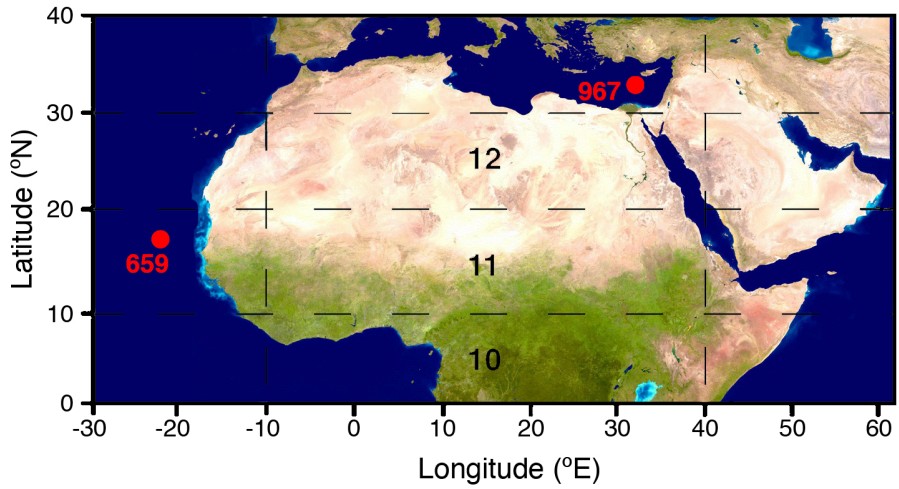

**Figure 1.** Map of the study area, with the background showing vegetation cover. Dashed lines indicate the grid boxes of CLIMBER-2, with the number the grid boxes used in this study. Red dots indicate the location of the sites.

computational costs, the model is particularly useful to simulate transient climate variability (e.g. Tuenter et al., 2005). More recent work resulted in a 5 million years simulation (Stap et al., 2018), for which this CLIMBER -2 simulation was driven by orbital variations, ice-sheet change and atmospheric $CO_2$ concentrations (Laskar et al., 2004; van de Wal et al., 2011; de Boer et al., 2014; Stap et al., 2018). We will compare Northern African continental runoff reconstructions of CLIMBER-2 with dust records associated to the North African monsoon (ODP Sites 659 (Tiedemann et al., 1994) and 967 (Lourens et al., 2001)).

For this purpose we will show for the first time the extended dust record of ODP Site 967 (Wehausen and Brumsack, 2000; Lourens et al., 2001) down to 3.2 Ma.

## 2    Methodology

### 2.1    CLIMBER-2 description

The CLIMBER-2 (CLIMate-BiosphERe model) is an Earth system climate model of intermediate complexity (Petoukhov et al.,

2000). The model contains simplified governing equations for processes and feedback systems of the atmosphere, oceans, sea ice and terrestrial vegetation. This allows the model to be used for global reconstructions over very long time scales. The CLIMBER-2 model has a low spatial resolution, where the longitude is divided into seven sectors each consisting of approximately $51°$ longitude, and the latitude into steps of $10°$ each (Figure 1). The land-ocean fraction for each grid box varies accordingly to the shape of each continent. The CLIMBER-2 model has a temporal resolution of one day, which are

averaged to capture annual averaged output. For the analysis we have used 1-kyr running mean output of CLIMBER-2.

The atmosphere component of CLIMBER-2 shares many features with more sophisticated models (Petoukhov et al., 2000). The model comprises of vertically averaged prognostic equations to model temperature and water vapour, and to reconstruct horizontal transport and radiative fluxes. Specific humidity is assumed to be exponential in the vertical profile, and the wind velocity is based on both geostrophic and ageostrophic components. It is assumed that the Hadley, Ferrel and polar cells are robust, and must also have existed under different climatic conditions. Within CLIMBER-2 six surface types can exists: sea ice or open water for ocean, and ice sheets, grassland, forest, and desert for land. These are all calculated separately and multiple types can coexist within one grid box. For example, grid box 11 and 12 are completely land boxes, but grid box 10 is 40% ocean and 60% land (Figure 1). CLIMBER-2 further includes a three-basin ocean model based on (Stocker et al., 1992). It describes the zonally averaged flow of water, as well as the evolution of temperature and salinity in the Atlantic, Indian, and Pacific basins, which are connected through the Southern Ocean. It consists of 20 uneven levels and an upper mixed layer of 50 meters, using a time step of 5 days and a latitudinal resolution of 2.5°.

The vegetation module VECODE is a dynamical global vegetation model that is based on the classification by Brovkin et al. (1997), who provided a continuous bioclimatic classification. This component describes the spatial behaviour of vegetation and its corresponding carbon fluxes. It is assumed that vegetation cover is in equilibrium with climate. Therefore, changes in vegetation will affect the albedo, roughness length, and transpiration of the grid box, resulting into a changing climate. The temporal resolution of the vegetation module is one year. The planetary albedo of a grid box is calculated based on the amount of snow- and vegetation cover, as well as the degree of cloudiness for every grid box. A two-layer soil model is also incorporated, in which the upper soil layer is determined by precipitation, evaporation, and transpiration, the melting of snow, and runoff and drainage. On the contrary, runoff and precipitation also depends on the water content and amount of vegetation that grows on the upper soil layer.

## 2.2 Setup of CLIMBER-2 experiments

This study focuses specifically on the Northern African continent (Figure 1), for which we have used output of a 5 million year simulations of CLIMBER-2. To achieve such a long simulation, the model is forced with changes in orbital parameters, atmospheric $CO_2$ and ice sheets (Stap et al., 2018). For the orbital variations the solution from Laskar et al. (2004) is used, describing variations of eccentricity, climatic precession and obliquity (Figure 2a,b). Ice volume reconstructions are enforced on land surface points of CLIMBER-2 influencing albedo and surface elevation. The ice sheets are taken from a global simulation by de Boer et al. (2014), who calculated global ice volume changes based on the LR04 benthic $\delta^{18}O$ stack by Lisiecki and Raymo (2005), shown here in Figure 2c. They used an inverse modelling approach to derive ice volume (Figure 2d) and surface-air and deep-ocean temperature. For the CLIMBER-2 model, an updated version of the proxy data composition by van de Wal et al. (2011) is used for $CO_2$ concentrations (Stap et al., 2018). To reconstruct $CO_2$ variations (Figure 2e) the same approach was used as in van de Wal et al. (2011), by calibrating $CO_2$ records against global temperature change. Instead the calibration here is based on the temperature simulation associated with the ice sheet simulation also used in the CLIMBER-2 model. Other variations in greenhouse gases are not taken into account in the model. This way the simulation of CLIMBER-2

is consistent with the forcing, driven by Laskar et al. (2004) insolation and climatic forcing records on the LR04 age scale (Lisiecki and Raymo, 2005).

Using the three different climatic records, four different climate model simulations are performed using only i) orbital variations (indicated with 'O'), ii) orbital and the greenhouse gas $CO_2$ (OG), iii) orbital and ice sheets (OI) and iv) all records (OIG). These four different simulations are used here as well to analyse variations in African runoff. When forcing records are kept constant we use the present-day ice sheets and a pre-industrial level of 280 ppm for $CO_2$.

## 2.3 Climatic dust records

We will compare the output of CLIMBER-2 with three different sites presenting dust or terrestrial variations driven by climate variability over the African continent (Figure 1). The core drilled at ODP Site 967 is an exceptional record of paleoclimate with detailed cyclic variability (Lourens et al., 2001). It is located in the eastern Mediterranean Sea, south of Cyprus. The interval consists of six sapropels, also analysed through colour reflectance (Lourens et al., 2001). The sapropels are cyclostrati-graphically correlated to the sections of Hilgen (1991), Lourens et al. (1996), and Kroon et al. (1998), which all originate from different sites within the Mediterranean. The Ti/Al ratio proxy is used to reconstruct variations in climate, whereby variability in Ti/Al represents variation in relative contribution of aeolian (i.e. Ti-rich dust particles) and fluvial (i.e. Al-rich suspended clay components) terrigeneous input in the sediment core (Wehausen and Brumsack, 2000; Lourens et al., 2001; Konijnendijk et al., 2014; Grant et al., 2017). It is suggested that Sahara dust is a prior contributor to the aeolian flux, while a large propor-tion of the fluvial input originates from the river Nile. Hence, low Ti/Al values indicate humid conditions, for example during periods of sapropel formation in the eastern Mediterranean, and high Ti/Al values correspond to more arid conditions at the northern part of the African continent (Wehausen and Brumsack, 2000; Lourens et al., 2001). An astronomically-tuned Ti/Al time series was established for the 2.9-2.4 Ma time interval by correlating minimum (maximum) values in the Ti/Al record to their representative maximum (minimum) values in the Laskar et al. (2004) summer insolation curve at 65°N latitude (Lourens et al., 2001). Here, we extend the tuned Ti/Al time series down to 3.2 Ma, using up to now unpublished Ti/Al data that has been generated using similar procedures as described in Lourens et al. (2001). All samples were freeze-dried and then ground and homogenised in an agate ball mill. For XRF-analysis, 600 mg of the sample powder were mixed with 3,600 mg lithium etrabo-rate ($Li_2B_4O_7$, Spektromelt), pre-oxidised at 500 °C with $NH_4NO_3$ and fused to glass-beads that were subsequently analyzed with a Philips PW 2400 X-ray spectrometer. Analytical precision was determined by parallel analysis of one international (GSR-6) and several in-house standards and was better than 1% for Al and Ti.

ODP Site 659 is located on top of the Cape Verde Plateau, northwest of Africa (Figure 1). Its dust record encompasses the past 5 Myr, and is mostly influenced by the African easterly jet stream, which transports dust from the Sahara-Sahel region towards the Atlantic Ocean (Tiedemann et al., 1994). We have re-tuned the dust record of ODP Site 659 (Wang et al., 2010) to the LR04 benthic stable isotope stack (Lisiecki and Raymo, 2005). As mentioned in Wang et al. (2010), the data for the last 2.6 Myr follow the same age scale, which is actually included in the LR04 stack (for benthic $\delta^{18}O$). Between 5.2 and 2.6 Ma the data is retuned to minima in the Laskar et al. (2004) 65°N insolation curve.

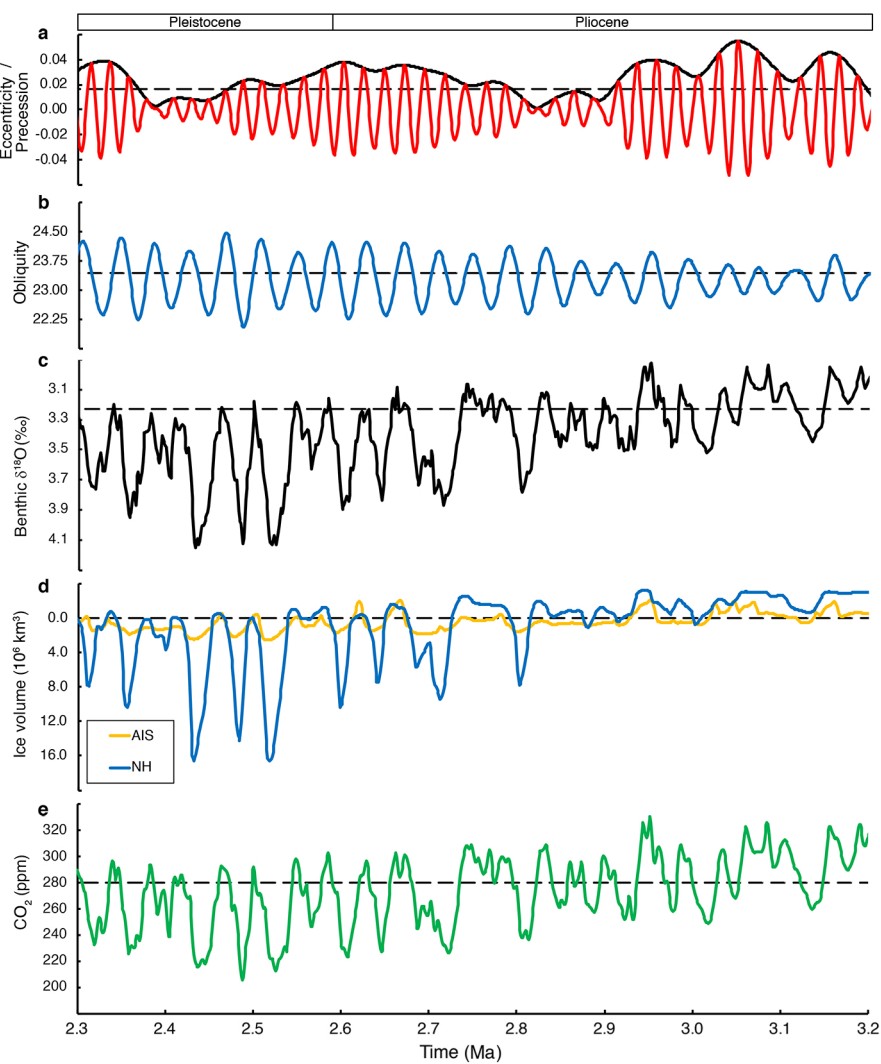

**Figure 2.** Climatic forcing records used as input for CLIMBER-2 over the time studies from 3.2 to 2.3 Ma. a) Eccentricity in black, climatic precession in red and b) obliquity in blue (Laskar et al., 2004). c) The benthic LR04 $\delta^{18}$O record (Lisiecki and Raymo, 2005) used for reference. d) Global ice volume reconstruction (in $10^6$ km$^3$ on land) for the Antarctic ice sheet (AIS) in orange and the total NH in blue from de Boer et al. (2014). The y-axis are reversed for c and d, with cold climate below. e) Atmospheric $CO_2$ concentrations (in parts per million; ppm) generated using the methodology described in van de Wal et al. (2011).

## 3  Orbital pacing of CLIMBER-2 modelled runoff

Runoff over the Northern African continent as modelled in CLIMBER-2 largely results from the difference between precipitation and evaporation over land, although water can also be stored in the soil, lakes and groundwater. As can be seen in Figure 1, 145 grid box 12 is mainly covering the Sahara desert, whereas grid box 11 includes the Sahel region, which is more dominated

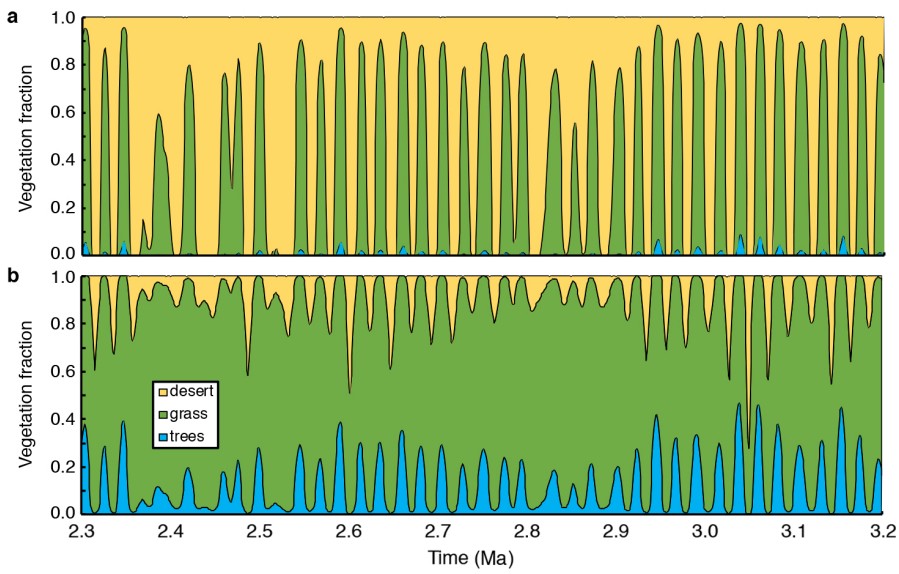

**Figure 3.** OIG Vegetation fractions over a) grid box 12 (Sahara) and b) grid box 11 (Sahel). Fraction of desert is given in yellow, grass is green and trees is blue.

by grassland. The variations of vegetation coverage for both grid boxes over time shows a similar patterns as illustrated in Figure 3. Clearly, during the studied time interval from 3.2 to 2.3 Ma the Sahara region (Figure 3a) is dominated by desert, which is largely replaced with grassland when precipitation is high. Although the Sahel region (Figure 3b) is largely covered with grassland, it is partially covered with trees and a reduction in desert during precession minima. Both grid boxes show high variability of the vegetation fraction, but do show a clear linkage to the present-day coverage of partially desert and vegetation for grid box 11 and desert for grid box 12 (Figure 1).

The difference between the two grid boxes is clearly represented by the variation in runoff over the Plio-Pleistocene transition (Figure 4a). It is evident that the minimum runoff of grid box 11 is approximately the equivalent of the maximum runoff of grid box 12. Grid box 11 has large variations in amplitude, with runoff maxima occurring during precession minima (Figure 4a). For the Sahel region, the runoff is strengthened following the African summer monsoon, driven by an increase in NH insolation. The increase in precipitation causes an increase in trees, which replaces desert and grass (Figure 3b). This enhances evaporation, which is stronger than for grassland, causing the peaks in runoff. In contrast, the runoff values of grid box 12 (Sahara desert) do not increase by the strengthened monsoon, but show peaks of low runoff during precession maxima. Although precipitation is enhanced during the summer monsoon when the air from the Atlantic Ocean reaches land, higher temperatures provide more room for water to evaporate, in combination with an increase of grass cover (Figure 3a) In the case of grid box 12, this additional precipitation is therefore compensated by an increase in evaporation. Also, during precession maxima precipitation is reduced and vegetation disappears, which leads to a strong decrease of evaporation and minima in the runoff.

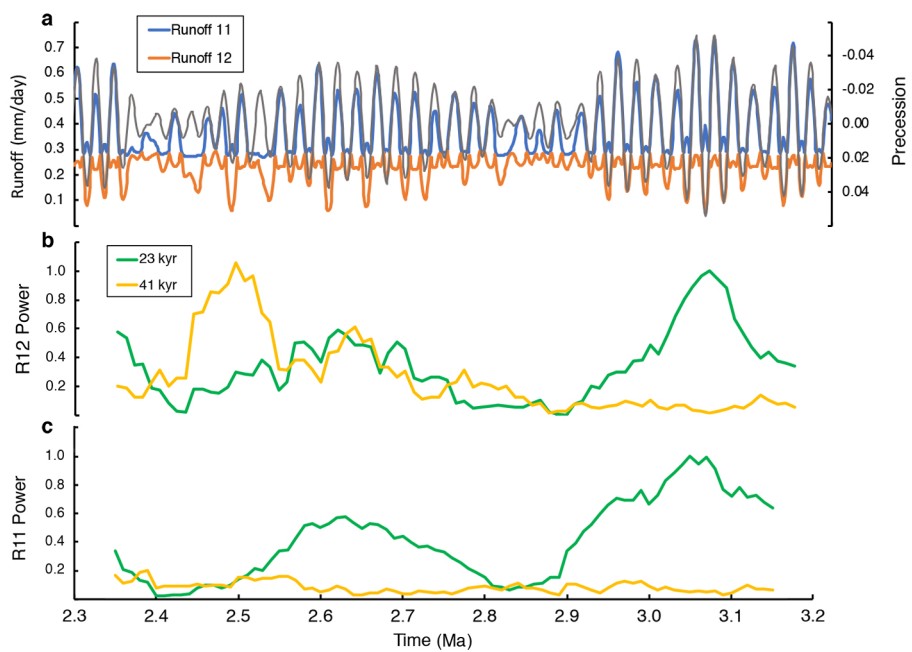

**Figure 4.** OIG Runoff over grid box 11 (blue) and 12 (orange), as given in Figure 1 in mm per day. The background shows climatic precession (Laskar et al., 2004) with the scale given on the right y-axis that is reversed (grey). Evolutive normalised power spectrum is given for b) grid box 12 and c) grid box 11, for 23 kyr (green) and 41 kyr (yellow), using a time window of 100 kyr.

The clear dominance of precession variability during this time interval is well depicted by the global power spectra of both regions (Figure 5a). Over time, the evolutive power spectra confirm the dominance of precession, although it is lower during
eccentricity minima (Figure 4b,c). Clearly high latitude influence is more present for the Sahara (grid box 12), showing an increased obliquity power after inception of NH glaciation (Figure 4b).

The influence of precession variability on African runoff on both regions is strongly present in all four CLIMBER-2 experiments (O, OG, OI and OIG) over the Plio-Pleistocene transition (Table 1). For all runs the lag to climatic precession is minimal, showing that for this particular frequency and interval, the modelled runoff is largely in sync with climatic precession. In the
case of obliquity, the lag is also small for the O and OG experiments, not including ice-sheet changes. However, when ice-sheet changes are included in the CLIMBER-2 OI and OIG simulations, we see an increase in the lag by about 1.5-2 kyr relative to the runs with constant ice-sheet topography. This shift can be attributed to the imposed lag in the tuning of the LR04 benthic $\delta^{18}O$ data when calibrating the depth-age scale. The time lag between obliquity (41-kyr) and its related frequency component in the LR04 stack is gradually increased from 3 kyr prior to 3 Ma towards 5-6 kyr up to 1.2 Ma. This follows from an antici-
pated slower response time of the growth of larger Pleistocene ice sheets (Lisiecki and Raymo, 2005). Also, the power of the obliquity frequency of runoff is increased in the OI and OIG relative to the O and OG simulations.

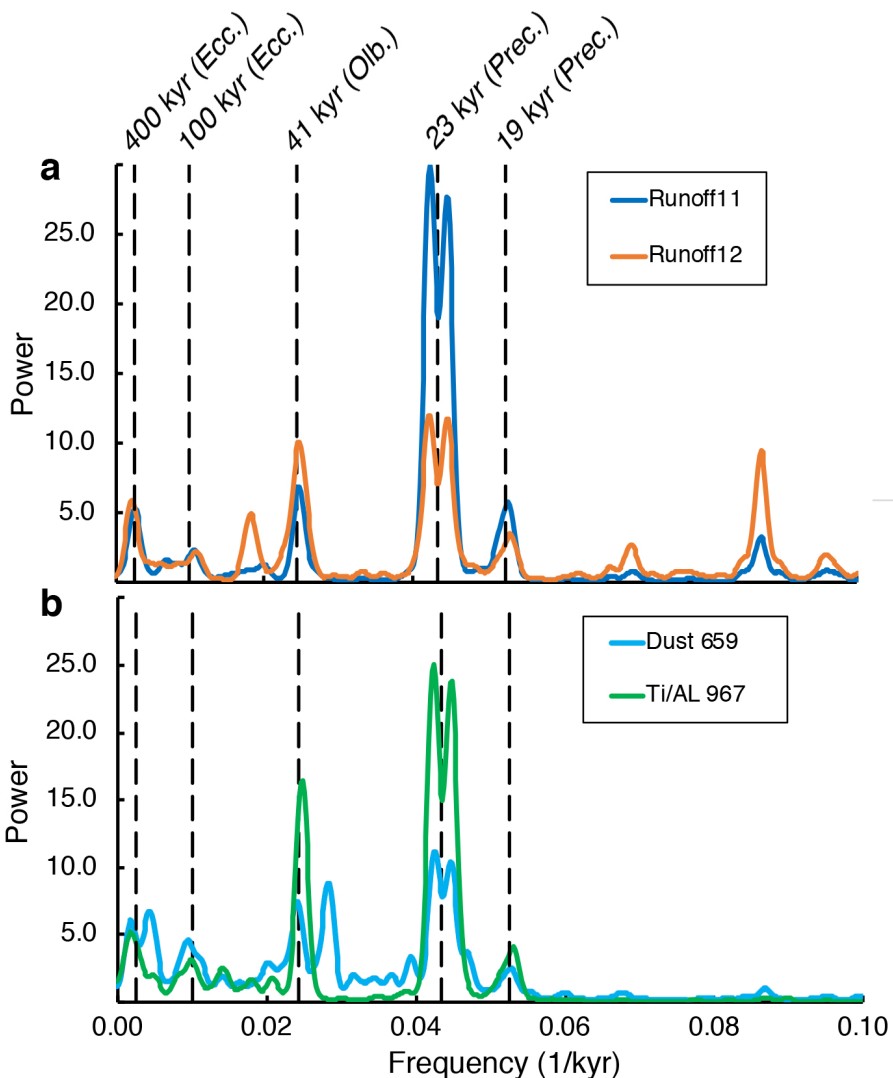

**Figure 5.** Normalised power spectra over the period 3.2 to 2.3 Ma of a) runoff of grid box 11 (blue) and 12 (orange), and b) of Ti/Al of Site 967 (green) and dust of Site 659 (light blue). Site 659 dust record is re-tuned to the LR04 benthic stable isotope stack (Lisiecki and Raymo, 2005). Power spectra are created with AnalySeries (Paillard et al., 1996), using a Blackman-Tukey spectral analysis with a Parzen window with 90% lags. Vertical dashed lines indicate the dominant orbital periods of 400 and 100 kyr (eccentricity), 41 kyr (obliquity) and 23 and 19 kyr (precession).

## 4    Comparison of North African runoff with offshore sediment records

The orbital periodicity is presented by the power spectra of Sites 967 and 659 (Figure 5b). A strong presence of precession and obliquity is seen for both the Ti/Al record of Site 967 (Lourens et al., 2001) and the dust record on the re-tuned age model

**Table 1.** Calculated time lags in kyrs between orbital frequencies and runoff over grid box 11 and 12 for the full period 3.2 to 2.3 Ma. Frequencies are extracted from the data with a Gaussian filter for precession at 23 kyr ($0.0435 \pm 0.003$) and for obliquity at 41 kyr ($0.0245 \pm 0.003$). Data analysed with AnalySeries (Paillard et al., 1996), using a Blackman-Tukey spectral analysis with a Parzen window with 90% lags. CLIMBER-2 experiments are O: orbital, OG: orbital and $CO_2$, OI: orbital and ice sheets, OIG: orbital, $CO_2$ and ice sheets.

| Period | R11-O | R11-OG | R11-OI | R11-OIG | R12-O | R12-OG | R12-OI | R12-OIG |
|---|---|---|---|---|---|---|---|---|
| Precession (23 kyr) | 0.194 | 0.221 | 0.230 | 0.248 | -0.188 | -0.163 | 0.184 | 0.228 |
| Obliquity (41 kyr) | 0.728 | 1.135 | 2.697 | 2.604 | 0.497 | 0.772 | 2.390 | 2.667 |

**Table 2.** Correlation of runoff from grid box 11 and 12 with the sedimentary records at Site 967 and 659 Correlation is performed over the full period 3.2 to 2.3 Ma, all data is interpolated on a 1 kyr time step.

| Region | Ti/Al 967 | Dust 659 |
|---|---|---|
| Runoff 12 (Sahara) | 0.586 | 0.560 |
| Runoff 11 (Sahel) | 0.662 | 0.361 |

of Site 659. We have also analysed the coherence and phase lag for the precession frequencies from the runoff and the dust records. For this we have filtered the precession frequency of $1/23 \text{ kyr}^{-1}$ ($0.0435 \pm 0.003 \text{ kyr}^{-1}$) for each record and compared it relative to precession during the full time interval from 3.2 to 2.3 Ma. Coherence is high for Runoff from grid box 11 and 12 and for the Ti/Al record (all above 0.99), with a phase lag of about 200 years. For the re-tuned dust record of Site 659 the coherence is 0.95 and the phase lag is $\sim$800 years. The correlation with the runoff output from CLIMBER-2 is strongest with the Ti/Al record from the Mediterranean (Table 2). The dust record from Site 659 shows on the contrary a good correlation with runoff from box 12 (Sahara) and a much weaker correlation with box 11 (Sahel). Note that for the correlation we have used the inverse relationship, correlating high runoff with low dust output, and corrected for the time lags of 1 kyr for the chronologies of ODP Site 659.

### 4.1 North African runoff and sapropel formation in the Mediterranean

The influence of the North African monsoon and its relation with orbital forcing is also clear through the presence of Mediter-ranean sapropels (e.g. Hilgen, 1991; Lourens et al., 2001). Sapropels are organic-rich sedimentary layers, resulting from anoxic events caused by an increase in runoff that is associated with a strong summer monsoon. Over the Plio-Pleistocene transition we compare the Ti/Al record with CLIMBER-2 runoff form grid box 11 and 12 (Figure 6). Clearly, both the wet- and dry periods correspond well with the simulated runoff, although the runoff maxima from grid box 11 (Sahel) largely link to low Ti/Al peaks, and runoff minima of grid box 12 (Sahara) overlay with maxima of the Ti/Al record. The sapropel layers that have been found in ODP Site 967 (Emeis et al., 1996; Kroon et al., 1998; Lourens et al., 2001) can be correlated with the high runoff peaks of grid box 11. This shows that the Sahel region does not only indicate the increased African monsoon phases,

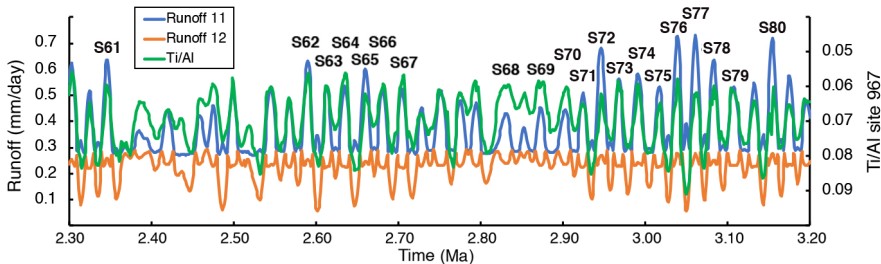

**Figure 6.** A comparison between CLIMBER-2 runoff from grid box 11 (blue) and 12 (orange) and the Ti/Al record from Site 967 (green) from 3.2 to 2.3 Ma. Sapropel layers are labeled (S61 to S80) as adopted from Kroon et al. (1998) and Lourens et al. (2001) corresponding to Ti/Al minima and runoff maxima. Note that the left y-axis for the Ti/Al is reversed, with wet phases on top.

but also correlates to the corresponding sapropel layers in the Mediterranean Sea. The Sahara region correlates with the high Ti/Al values, indicating dry phases. These dry phases of the Sahara region can therefore be correlated to the marls, i.e. periods
of high dust flux, in between the sapropel layers of ODP Site 967 Lourens et al. (2001).

Although the general pattern correlates very well (Table 2), there are also clear differences between the CLIMBER-2 simulations and the Ti/Al record. For example, the relative amplitude variations of the humid phases are not always the same. Sapropel 64, is clearly enhanced by obliquity, whereas the runoff at this time has a relatively low maximum. On the other hand, runoff is significantly high during the formation of sapropel 65, which is one of the less distinct sapropel layers. This is remarkable,
as a large runoff in Northern Africa would suggest a lower Ti/Al ratio due to more riverine input in the Mediterranean Sea. The most apparent discrepancy between the Ti/Al record and the CLIMBER-2 simulated runoff is shown during the 400-kyr related eccentricity minimum around 2.8 and 2.4 Ma. During these long-term eccentricity nodes, the Ti/Al record suggests on average drier climate conditions in northern Africa than indicated by the model simulations At this time interval, both the runoff amplitudes are significantly lower and the variations for grid box 12 are even out of phase with the sapropel layers S68
and S69 (Figure 6).

## 4.2 Combined runoff and overall correlation

It appears that the influence of wetter periods (grid box 11) overshadows the effect of dry periods (grid box 12), given that the amplitude variations of grid box 12 (Sahara) are a lot less than runoff from grid box 11 (Sahel). Both regions include the catchment area of the river Nile (e.g. Bosmans et al., 2015b), and hence a combination of runoff of the two sites would be
more suited for a comparison with the Ti/Al record. We have investigated multiple combinations of runoff from grid box 11 and 12 and compared these with the two sedimentary records of Sites 967 and 659 over the full period from 3.2 to 2.3 Ma (Figure 7). For each grid box, we have used a scaling varying between $0.0$ and $1.0$, in order to cover all possible combinations of the two records, including the individual correlation. For both the Mediterranean (Site 967) and the Atlantic Ocean (Site 659) a combined runoff record provides the best correlation with the data.

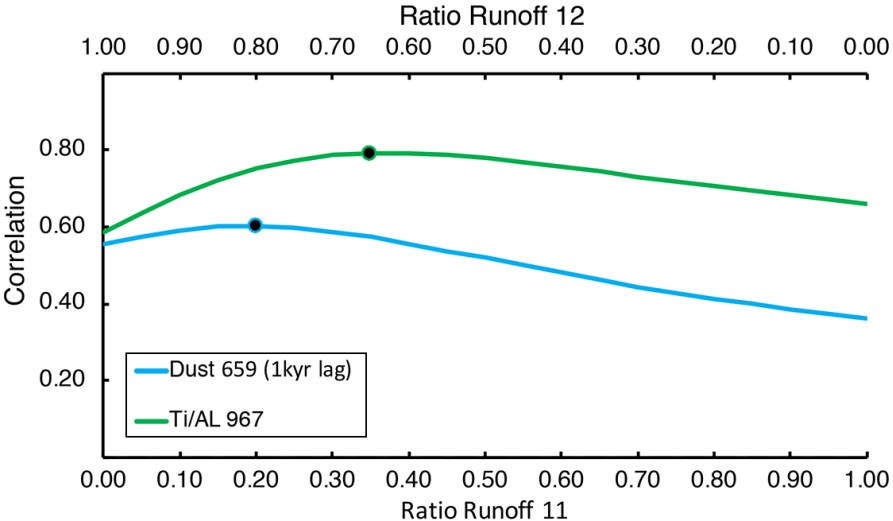

**Figure 7.** Correlation of combined output of runoff from grid box 12 (top x-axis) and 11 (lower x-axis). For example, 0.5 and 0.5 means a runoff record consisting of 50% runoff 11 and 50% runoff 12. For each created record the correlation is computed over the full time period of 3.2 to 2.3 Ma. In green with TI/Al of Site 967, in light blue with dust from Site 659 (lag of 1 kyr). The dots indicate the maximum correlation: for Ti/Al the record of 0.35R11 + 0.65R12 (0.792) and for Site 659: 0.2R11 + 0.8R12 (0.602).

The combination of 35% runoff 11 plus 65% runoff 12 has a correlation of 0.792 with the Ti/Al record, and 20% runoff 11 plus 80% runoff 12 has a correlation of 0.602 with the dust record of Site 659. For both sites this correlation is higher than with the individual grid boxes, as shown in Table 2 and Figure 7. The correspondence between the combined runoff record and the Ti/Al data is high throughout the Plio-Pleistocene time interval (Figure 8a). For separate time windows of 100 kyr the correlation is consistently high, with a maximum value of 0.930 between 2.6 and 2.5 Ma, and a minimum value of 0.749

between 2.5 and 2.4 Ma. For the dust record of Site 659 the correlation is less pronounced for the individual records (Table 2). When we analyse the 100-kyr time window the correlation with the combined runoff record (Figure 8b) varies between -0.103 and 0.873, with only shows a negative correlation between 2.9 and 2.8 Ma and an overall high correlation between 0.524 for 2.7 - 2.6 Ma and 0.873 for 2.8 to 2.7 Ma.

Although correlation is high for some time intervals, there is a non-linear behaviour between runoff, which results from pre-

cipitation and evaporation, and dust peaks from Site 967 and Ti/Al from Site 659. High runoff peaks do not always correspond to high dust or Ti/Al signatures in the records. We have illustrated this by comparing the high and low peaks of Ti/Al and the dust records with the corresponding peaks of the runoff records (Figure 8c,d). For both the high (orange) and low (light blue) peaks of Ti/Al (Figure 8c) a clear trend is visible. On the contrary, for the dust record (Figure 8d), the high peaks (red) show a more linear trend compared to the low peaks (blue). Nonetheless, correlation coefficients are moderate too low for all

comparisons.

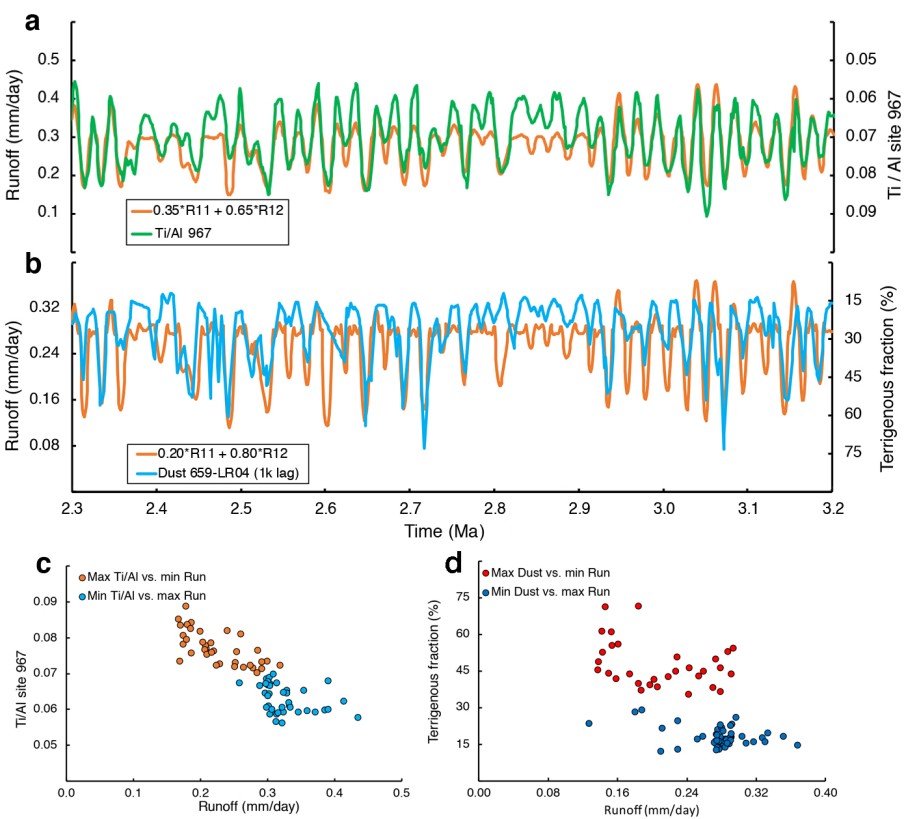

**Figure 8.** A comparison between CLIMBER-2 combined runoff output, that correlates the highest as given in Figure 7. a) Combined runoff of 35% runoff 11 plus 65% runoff 12 (orange) and the Ti/Al record of Site 967 (green). b) Combined runoff of 20% runoff 11 plus 80% runoff 12 (orange) and the dust record of Site 659 (light blue) re-tuned to the LR04 age scale. In a and b, for both sedimentary records the y-axis is given on the right and is reversed. c) A comparison of the high (orange) and low (light blue) peaks of Ti/Al versus runoff from panel a. peaks are selected for Ti/Al values above and below 0.07. d) A comparison of the high (red) and low (blue) peaks of Dust versus runoff from panel b. peaks are selected for Dust values above and below 35.

Besides the good correlation during most 100-kyr intervals also the evolutive power of especially the precession frequency (blue curve in Figure 9a,b) shows a strong similarity with the precession power in CLIMBER-2 runoff (Figure 4b,c). A clear lowering is seen for the eccentricity minimum from 2.8 to 2.9 Ma. For the dust record of Site 659, besides the eccentricity minima a dip in the 23-kyr power around ~2.6 Ma is seen (Figure 9b), for which we would expect 23-kyr power to stay high
following the 400-kyr eccentricity maximum (Figure 2a). This also underlines the non-linear link between dust and runoff as illustrated in Figure 8.

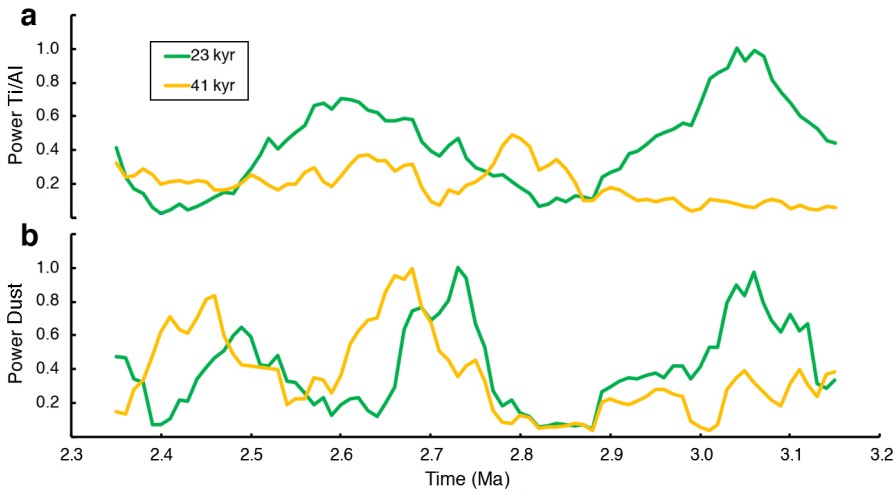

**Figure 9.** Evolutive normalised power spectrum for a) Ti/Al of Site 967 and b) Dust of Site 659, for 23 kyr (green) and 41 kyr (yellow), using a time window of 100 kyr.

## 4.3 Transient impact of runoff to the Mediterranean

The strong link between simulated runoff and the Ti/Al record of the Mediterranean is a feature that is very persistent through the Plio-Pleistocene transition (Figure 8a). Although the strongest correlation over the whole period is found for the combi-
nation of 35% runoff 11 plus 65% runoff 12, correlation is generally high for other combinations as well. This provides us an opportunity to not only look at the temporal correlation of individual sites, but also how the contributions of runoff from the regions impacts the Ti/Al record over time. Figure 10a presents this for 100-kyr time windows, showing for each window separately the combination of runoff from grid box 11 and 12 with the highest correlation (illustrated in Figure 10b by the grey line), depicting the transient relationship of runoff over Northern Africa to sediment deposition in the Mediterranean.

There is a clear transition present, coinciding with the increase of NH glaciation (Figure 2d), at around 2.8 Ma. Before this time, the correlation is highest for a more dominant contribution of the Sahara region (grid box 12 in orange), close to the overall coherence with 35% of grid box 11 and 65% grid box 12. After 2.8 Ma, the highest correlation is found for a more equal contribution of 50% each of the two regions. We found no strong trends in runoff of both regions.

The catchment area of the river Nile stretches more south than regions 12 and 11, starting around lake Victoria (the blue
lake in Figure 1 at the equator, 33°E). Therefore, we also checked correlation with runoff from region 10, using combinations with runoff from regions 11 and 12. Surprisingly, the individual correlation of runoff from region 10 with the Ti/Al record from 3.2 to 2.3 Ma is higher (0.720) than that of regions 11 and 12 (Table 2). Before 2.8 Ma, correlation is highest for different combinations of all three regions, but after 2.8 Ma again a 50% contribution from regions 11 and 12 is strongest, with no contribution from region 10. Since region 10 is (far) more south and largely contains the African tropical forest, drainage from

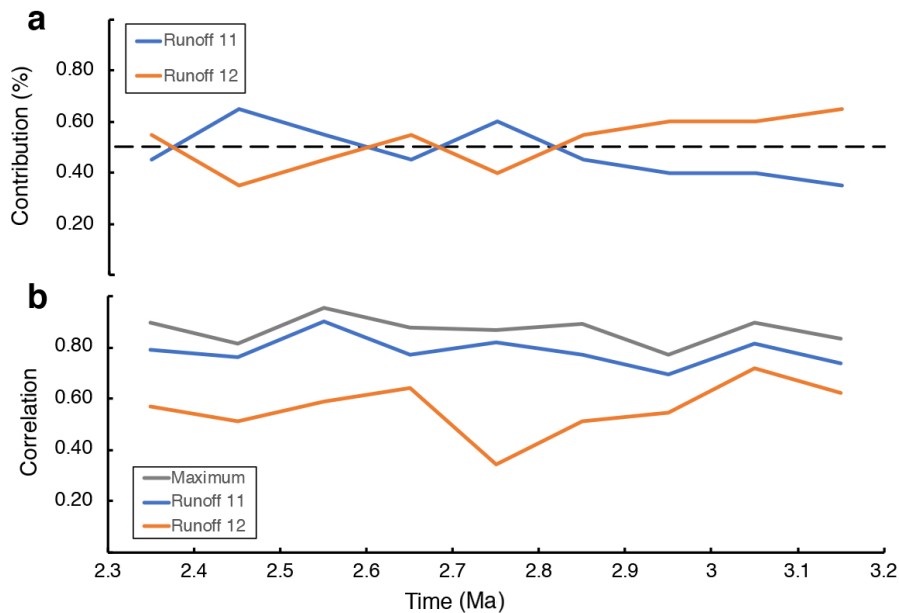

**Figure 10.** Transient contribution of runoff from grid box 11 and 12 to Mediterranean dust input. For each time window of 100 kyr, all combinations of the two grid boxes (as in Figure 7) are correlated with the Ti/Al record of Site 967. a) The contribution of both grid boxes, blue for runoff 11 and orange for runoff 12, is given corresponding to the maximum correlation within that time frame (grey line in panel b). b) the maximum correlation within each 100-kyr time window is given in grey, corresponding to the contributions shown in panel a. Correlation of the individual grid boxes is given in blue for runoff 11 and in orange for runoff 12.

central Africa is also towards the Atlantic Ocean through the Congo river, which might clarify the lower influence of region 10 when combining the regions.

## 5   Discussion and conclusions

In this paper we have used simulations of the climate model CLIMBER-2, and analysed reconstructed climate variables from 3.2 to 2.3 Ma, a period that includes the Plio-Pleistocene transition. The model has been used previously to link orbital and

climatic variability and is well suited for long-term transient climate simulations (e.g. Tuenter et al., 2005). During this time interval, the world experienced a large increase in NH glaciation, concurrent with a draw down in atmospheric $CO_2$ concentrations (Figure 2). The CLIMBER-2 simulations have been run over 5 million years, and included ice-sheet changes, atmospheric $CO_2$ variations and orbital parameters (Stap et al., 2018). We have looked at climate variability over the Northern African continent and have linked transient variations of continental runoff to sedimentary records in the Mediterranean Sea and Atlantic

Ocean.

The climatic variability in the model is largely determined by changes in the forcing records; NH and Antarctic ice sheets, atmospheric greenhouse gas forcing of $CO_2$, and orbital variations (Laskar et al., 2004). Both timing and magnitude of the

forcing will have an impact on the changes shown by the model. The ice-sheet forcing imposed here is based on a 3-D ice-sheet model constrained by the LR04 benthic $\delta^{18}$O stack by Lisiecki and Raymo (2005), which also determined the age scale of the $CO_2$ reconstructions. Ice-sheet changes can be different, but are constrained by the locations. On the other hand, $CO_2$ is much less constrained since proxy data over this time period are sparse, but do reflect an amplitude of about 40-100 ppm (Martinez-Boti et al., 2015; de la Vega et al., 2020), and model-based reconstructions can be quite different (see for example Figure 6 in Berends et al., 2020). Particularly, the reconstruction from Stap et al. (2016) shows a much larger amplitude in $CO_2$ (80-150 ppm), whereas that from Willeit et al. (2019) employs a smaller amplitude (40-50 ppm) over our time period compared to our forcing record (60-80 ppm).

Simulated runoff over the Northern African continent shows periodic behaviour largely related to the orbital frequencies of precession, although obliquity influence is also present, especially after inception of NH ice (∼2.8 Ma). Previous studies have shown the strong presence of orbital induced variations of the African monsoon, that could originate from enhanced moisture transport from the tropical Atlantic (Bosmans et al., 2015a). From 3.2 to 2.3 Ma the variations are largely in sync with climatic precession. Obliquity variations are much less pronounced but do show an increase in the lag after inception of NH ice sheets at about 2.8 Ma. This is also related to the induced time-lag increase in the LR04 age scale (Lisiecki and Raymo, 2005), which is the origin of the ice-sheet and $CO_2$ reconstructions used in the CLIMBER-2 simulations (de Boer et al., 2014; Stap et al., 2018).

Although the evolution and dispersal of hominin species during this time could be linked to orbital variations (Joordens et al., 2019), a direct link with the climatic variations shown by CLIMBER-2 cannot directly be established.

The runoff output of Northern Africa correlate exceptionally well with the Ti/Al record of ODP Site 967, resulting in the correlation of sapropels S61-S80 to the corresponding wet runoff phases (Figure 6). Although the correlation is high, peak values of T/Al do not correspond exactly with low runoff, illustrating the non-linearity in the system that is for example depending on wind transportation of aeolian dust, and variable river flux of the Nile. Correlation with the dust record of ODP Site 659, which we re-tuned to the LR04 age scale, is moderate compared to that with the Ti/Al record. The lower correlation with the dust could be expected since dust emissions are a strongly non-linear function of ground cover, wind and soil moisture (e.g. Bauer and Ganopolski, 2010). Moreover, the hydrological cycle over the grid boxes is also clearly linked to the vegetation (Figure 3). The vegetation shows a high correlation for trees and desert with Ti/AL for the Sahel region (-0.734 and 0.781, respectively), and grass and desert for the Sahara region (-0.783 and 0.783). On the contrary, the correlations between vegetation coverage over the two regions is generally poor compared to the dust of Site 659, illustrating the strong non-linear relationship between vegetation and dust outside of the African continent.

The Ti/Al record of the Mediterranean represents variation in relative contribution of aeolian and fluvial dust input in the sediment core, relating high continental runoff from the African continent to lower values of the Ti/Al record. Although overall correlation is best represented by runoff from the Sahel region, we found that a combined runoff from the Sahel (grid box 11) and the Sahara (grid box 12) gives the highest correlation with the record. Henceforth, a high fluvial input corresponds well with high runoff from the Sahel region, whereas a more aeolian input corresponds to dry periods of the Sahara desert, corresponding to both low runoff and possible higher dust transport from the desert.

We correlated the combined runoff output of grid box 11 and 12 with the Ti/Al record. Over the entire period the correlation is highest for a record that combines 65% of grid box 12 (Sahara) with 35% of grid box 11 (Sahel) as presented in Figure 11b.

Although the data correlates fairly well with the wet-dry index reconstructed from the same Site 967 (Grant et al., 2017) (Figure 11a; correlation with Ti/Al is 0.49, with the combined runoff is 0.45), the wet-dry index, representing Northwest and East Africa climate, does show a much larger long-term (eccentricity) component not present in the TI/Al or runoff data. Furthermore, we have calculated a wavelet coherence diagram (Figure 11d) indicating coherence of frequencies between the combined runoff from CLIMBER-2 and the Ti/Al record of Site 967 (Figure 11b and c, respectively). The arrows indicate

that for precessional periods (∼23 kyr) the records vary largely in phase from 3.2 to 2.3 Ma. Moreover, obliquity change do increase slightly after 2.8 Ma.

We have shown that there is a clear uninterrupted impact of runoff on the Mediterranean, for which we showed that prior to 2.8 Ma a higher contribution from the Sahara region is required for a better correlation with the Ti/Al record. However, after inception of NH glaciation a 50% contribution of each of the regions represents the highest fit. It seems that prior to a more

high latitude influence of global climate, dry periods over the Sahara have more impact on the Ti/Al record, i.e. during relative warm climate of the Late Pliocene. After ∼2.8 Ma, the global cooling trend gives way to a more equal impact from the Sahel and the Sahara regions, whereas equatorial regions are much less linked to the variations seen in the Mediterranean. Moreover, it has already been shown previously that river runoff from the south, in this case largely dominated by the river Nile, is related to the strength of the North African monsoon (e.g. Bosmans et al., 2015b). There is a clear connection between greater parts of

Northern African climate and the Mediterranean. Although the CLIMBER-2 model is of low resolution, it also shows to have a strong coherence with sedimentary records from the Mediterranean especially for precessional frequencies.

*Author contributions.*   BdB and MP carried out the analysis, BdB wrote the paper, with contributions from all authors. All authors contributed equally to the discussion and interpretation of the results.

*Competing interests.*   The authors declare that they have no conflict of interest.

*Acknowledgements.*   B. de Boer is funded through a grant from the SCOR Corporate foundation for Science. This research was funded by NWO-ALW grant (project number 865.10.001) and Netherlands Earth System Science Centre Gravitation (Grant 024.002.001) to L. J. Lourens. We would like to thank Erik Tuenter for running the CLIMBER-2 simulations and making the data available. We would like to thank Hans Brumsack and Rolf Wehausen for making the data of Site 967 available.

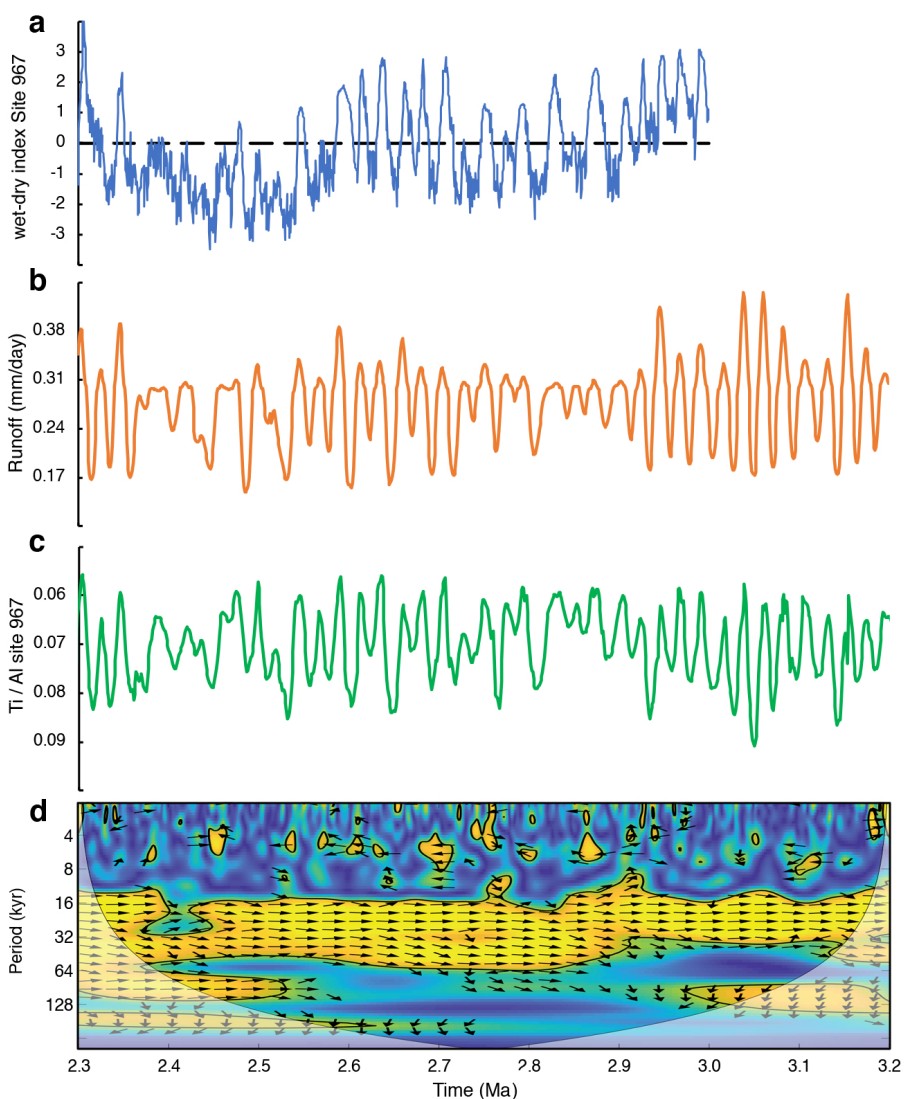

**Figure 11.** Records of African climate and Mediterranean sediments. a) The wet-dry index as calculated by Grant et al. (2017) from Site 967, indicating wet and dry phases of Northwest and East Africa. b) The optimal runoff combined record of 35% runoff 11 plus 65% runoff 12 from CLIMBER-2. c) The Ti/Al record from Site 967, with the y-axis reversed and d) a Wavelet coherence diagram (Grinsted et al., 2004) of Runoff (panel b) with Ti/Al (panel c). The colour scale indicates coherence, with >95% confidence levels indicated with the black lines (orange to yellow colours). The arrows indicate the phase, with right pointing meaning in phase, and upward pointing mean a lead of runoff relative to Ti/Al.

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
