# Peer review of "The transient impact of the African monsoon on Plio-Pleistocene Mediterranean sediments"

_Climate of the Past, 2020_

## Referee Comment (RC1) · Anonymous Referee #1 · 11 Sep 2020

In this paper the authors compare previously performed transient CLIMBER-2 climate model simulations with marine sedimentary records over the time interval from 3.2 to 2.3 Ma. This time interval includes the Pliocene-Pleistocene transition with the initiation of Northern Hemisphere glaciation. The analysis is focused on North Africa, in particular the Sahara and Sahel regions. Two sedimentary records, one in the Mediterranean (Ti/Al at ODP 967) and one in the Atlantic (dust at ODP 659), are compared with modelled runoff. The Ti/Al record is an extension to 3.2 Ma of an already published record from 2.4-2.9 Ma. The dust record of site 659 has been retuned for the purpose of this paper. The study shows that model and sedimentary records correlate relatively well and show a consistent response to orbital forcing, mainly precession. The last part of the paper is devoted to an analysis of the transient behavior of model-proxy relation.

[Figure]

The paper is of general interest for readers of Climate of the Past and fits into the scope of the journal, but I have a few major comments that need to be addressed before the paper is suitable for publication.

Major comments

My major concern about this paper is that it is at least partially based on a comparison between apples and pears. Runoff from the CLIMBER-2 model is compared to sediment records representing a combination of runoff and dust deposition (Ti/Al record of site ODP 967) and dust deposition (ODP 659). Since dust emissions are a strongly non-linear function of ground cover, wind and soil moisture, comparing runoff with dust deposition is hardly justified. They are of course related, during dry periods you expect less runoff and more dust, but their relation is probably far from linear. A direct comparison of runoff with the Ti/Al record is partly justified, because the Ti/Al record is expected to be also a proxy for runoff. This could also be the reason for the higher correlation of CLIMBER-2 runoff with the Ti/Al record compared to the correlation with the dust record at ODP 659. In principle the CLIMBER-2 model output probably includes all variables needed to diagnose the dust emission flux using e.g. the simple model described in Bauer & Ganopolski, 2010. This would allow a more straightforward comparison between model and the sedimentary records presented in the paper.

Because of its important effect on both the water cycle and dust emissions, I'm missing a description of what happens to the vegetation over the Sahara and Sahel in the model over the simulation period and how that could have affected runoff and dust and therefore the comparison with the sediment records.

A discussion of uncertainties in the forcings is missing. There are for example large uncertainties in the atmospheric $CO_2$ concentration. Proxy reconstructions show a large uncertainty, particularly in the amplitude of 'glacial-interglacial' $CO_2$ variability. The paper by Stap et al. 2016, just to name a model-based reconstruction where two of the authors of this paper are co-authors, shows a very different $CO_2$ trajectory

across the Pliocene-Pleistocene transition than that used in the simulations presented in this paper. I'm not saying that CLIMBER-2 should be re-run with all these alternative forcings, but a critical discussion of the possible impact that the choice of a particular forcing could have on the results presented in the paper is needed.

Minor comments

lines 27-29: What is meant by 'completely'? There are plenty of other studies that could be cited here, showing that, at least if $CO_2$ is low enough, orbital variations are enough to get pronounced glacial cycles: e.g. Abe-Ouchi et al., 2013 and Ganopolski & Calov 2011.

line 88: what does 'quality' mean here?

lines 104-106: sentence is unclear

lines 126-127: how has the tuning been done? Moreover that the LR04 stack has almost no precession for the early Pleistocene.

lines 136-139: Would be interesting to see the time series for precipitation, evaporation and runoff for the two grid cells. Also, what is happening to vegetation in these grid cells? Could it be that the increase in evaporation is related to an expansion of vegetation in the Sahara grid cell? If vegetation is growing over the Sahara I guess that more water should be available to evaporate because roots have access to deeper soil layers...?

lines 142-143: is this possibly related to changes in Atlantic meridional heat transport and subsequent changes in the position of the ITCZ when NH ice sheets start to grow and decay?

Fig. 3: Please mention in the caption that the y-axis for precession is reversed in 3a. It took me a while to figure out that maxima where actually minima.

lines 149-152: I have read this sentence 10 times, but still do not understand what it

means.

line 155: 're-tuned age model of Sites 659'. It is not a spectrum of the age model, but of the dust record, right?

line 254: 'we correlation combined': rewrite

line 257: 'which representing' -> representing

line 259: 'that indicating' -> indicating

Fig. 9d: and how are lags and leads represented? Please add a legend with arrow directions to clarify. Color scale is missing in d.

References

Stap, L. B., de Boer, B., Ziegler, M., Bintanja, R., Lourens, L. J. and van de Wal, R. S. W.: CO2 over the past 5 million years: Continuous simulation and new $\delta$11B-based proxy data, Earth Planet. Sci. Lett., 439(April), 1–10, doi:10.1016/j.epsl.2016.01.022, 2016. Bauer, E. and Ganopolski, A.: Aeolian dust modeling over the past four glacial cycles with CLIMBER-2, Glob. Planet. Change, 74(2), 49–60, doi:10.1016/j.gloplacha.2010.07.009, 2010. Abe-Ouchi, A., Saito, F., Kawamura, K., Raymo, M. E., Okuno, J., Takahashi, K. and Blatter, H.: Insolation-driven 100,000-year glacial cycles and hysteresis of ice-sheet volume., Nature, 500(7461), 190–3, doi:10.1038/nature12374, 2013. Ganopolski, A. and Calov, R.: The role of orbital forcing, carbon dioxide and regolith in 100 kyr glacial cycles, Clim. Past, 7(4), 1415–1425, doi:10.5194/cp-7-1415-2011, 2011.

---

## Referee Comment (RC2) · Anonymous Referee #2 · 24 Sep 2020

General Comments

The paper by de Boer and coauthors compares transient records of runoff from the Sahara and the Sahel that is produced by an intermediate complexity model with the dust record from 659 and the Ti/Al XRF record from 967 across the intensification of Northern Hemisphere glaciation. They find there these is strong precession-scale variability in all of the records, and that obliquity, thought to be driven by the high latitude ice sheets, has more of an impact on the Sahara following the intensification of NH glaciation. Authors also use a clever technique of correlating the empirical records with a range of combinations of grid cells to find the most likely combination of regional impact on the sediment and look at how this combination changes through time. The methodological steps are laid out in a fairly clear way with useful figures, but often

the details of the model and time series bog down the interpretation. There is little conclusion on what the lags, relationship between runoff and dust, and the change in regional input to the Mediterranean really means in terms of climate dynamics, and also what it means for humans evolving in East Africa. Please find my specific comments with more detail, as well as some technical comments on the text and figures, below.

Specific Comments

- There needs to be more background in the first paragraph of the introduction. There are newer studies that have good enough age constraints to examine the phase relationships between orbital properties and the MPT and other transitions. I think authors also need to make it clear that the 100 kyr cycle after the MPT is likely an average of 120 and 80, as many have recently shown. Further, the MPT is not studied in this research, so perhaps removing that from the introduction would leave more room to provide better detail on other arguments.

- There also need to be far more citations for a sentence like line 39-41, where the authors link orbitally induced variability in the tropics. This has been seen in Africa in many different records by Tierney, Lupien (both Pliocene and Pleistocene), Rose, etc. In the conclusions (line 239-241) there is a vague sentence on this as well – orbital induced variations of what? If authors are talking about precession in climate records from Africa, there need to be many more citations.

- Similarly, the third paragraph of the intro on hominin evolution is very light in citations and seems to pick specific details from Joordens 2019 rather than focusing on the evidence and mechanisms for hominin transitions at the onset of NH glaciation or the variability selection hypothesis (Potts).

- 'Continental runoff (i.e. p-e)' is a bit misleading. This p-e balance is not equivalent to runoff as water can be stored in soil, lakes, and groundwater, which is fully stated in the description of the model. This is brought up again in line 129 and onward and needs to be explained more clearly and thoroughly before the interpretations are stated.
- Figure 1 has the model grid cells, but they are not plotted over a map based on the 6 potential surfaces. This would be helpful.

- Why not use the principal component analysis, based on Ti/Al, from Grant 2017 for the 967 comparison? This is brought up later, and perhaps is a better estimate of climate than the raw Ti/Al record.

- Line 131: the location of the Sahel today is shown in Figure 1, but the location, spread, vegetation may well have been very different in the past, particularly in the Pliocene. How does this affect the grid cell coverage?

- The discussion surrounding the lags seems a bit tangential and perhaps unnecessary. If the lag is due to a choice in the LR04 tuning, then does this analysis tell us anything new?

- Line 207: the power spectra should not be influenced by lack of data. Either resample the data appropriately, or put the constraints up front in the methods section. Can authors resolve precession throughout the records? If not, then conclusions based on evolutive spectra shouldn't be made, or should be modified.

- What are the conclusions of the mechanisms linking the ice sheet inception to the change in runoff region source? The conclusions section appears to be more of a summary. Try to utilize this space for connections back to the topics brought up in the introduction – what about human evolution? Would this shift, and other aspects of the findings, impact humans, and how?

Technical Corrections

- "Myr age" can be replaced by "Ma" in every circumstance

- Line 42: replace 'deterioration'

- Line 42: 'central' Africa? Do authors mean East Africa?

- Figure 3: the colors need to match better for clarity. Perhaps in b, use orange with

different line types for the different frequencies, and in c, use blue.

- If table 1 could be shown in a figure, I think the point of lags would be much easier to comprehend by the reader. Perhaps use phase wheels.

- Figure 6: the legend should be changed to either show the location or the proxy, not the proxy for one site, and the site number for the other.

- Try not to conflate the Plio-Pleistocene transition with the onset of NH glaciation – authors have cited these at two distinct times (2.6 vs 2.8 Ma), so try to keep the wording consistent.

- There are multiple instances of awkward phrasing, run-on sentences, and misused gerunds, so further editing for grammar could benefit the clarity of the manuscript.

---

## Author Comment (AC1) · 16 Nov 2020

**Reply to the anonymous reviewer 1**

We would like to thank the reviewer for his/her comments. The remarks have definitely improved the manuscript. Below you will find a point-by-point reply, with the review given in **black** and our reply given in **blue**. We hope that we have answered all questions sufficiently. The line numbers we included in our answers refer to the marked changes document.

**Major comments**

My major concern about this paper is that it is at least partially based on a comparison between apples and pears. Runoff from the CLIMBER-2 model is compared to sediment records representing a combination of runoff and dust deposition (Ti/Al record of site ODP 967) and dust deposition (ODP 659). Since dust emissions are a strongly non-linear function of ground cover, wind and soil moisture, comparing runoff with dust deposition is hardly justified. They are of course related, during dry periods you expect less runoff and more dust, but their relation is probably far from linear. A direct comparison of runoff with the Ti/Al record is partly justified, because the Ti/Al record is expected to be also a proxy for runoff. This could also be the reason for the higher correlation of CLIMBER-2 runoff with the Ti/Al record compared to the correlation with the dust record at ODP 659. In principle the CLIMBER-2 model output probably includes all variables needed to diagnose the dust emission flux using e.g. the simple model described in Bauer & Ganopolski, 2010. This would allow a more straightforward comparison between model and the sedimentary records presented in the paper.

Non-linearity can clearly be seen when comparing peaks of Dust and Runoff. We thank the reviewer for putting this forward and have added this to the discussion of the results. We will discuss the non-linearity nature of the coupling in more detail, including referring to the Bauer & Ganopolski 2010 paper (thank you for the reference). We do not have the possibility to run the dust model ourselves, because not all CLIMBER-2 output is available to us at this stage. Added changes are:
at lines 232-238: "Although correlation is high for some time intervals, there is a non-linear behaviour between runoff, which results from precipitation and evaporation, and dust peaks from Site 967 and the Ti/Al record from Site 659. High runoff peaks do not always correspond to high dust or Ti/Al signatures in the records. We have illustrated this by comparing the high and low peaks of Ti/Al and the dust records with the corresponding peaks of the runoff records (Figure 8c,d). For both the high (orange) and low (blue) peaks of Ti/Al (Figure 8c) a clear trend is visible. On the contrary, for the dust record (Figure 8d), the high peaks (red) show a more linear trend compared to the low peaks (blue). Nonetheless, correlation coefficients are moderate too low for all comparisons."

And lines 287-289: "The lower correlation with the dust could be expected since dust emissions are a strongly non-linear function of ground cover, wind and soil moisture (e.g. Bauer and Ganopolski, 2010)."

Because of its important effect on both the water cycle and dust emissions, I'm missing a description of what happens to the vegetation over the Sahara and Sahel in the model over the simulation period and how that could have affected runoff and dust and therefore the comparison with the sediment records.

Yes, we understand. This was included in a first version of the manuscript, but the most outstanding result was the link between runoff and the sediment records. We of course agree that it is important to mention. We have included a new figure (now figure 3), which shows the vegetation fraction, either grass, trees or desert, over the two grid boxes. A short discussion is included in Section 3 (line 137 - 141) and the vegetation fractions are now included in the discussion of the results (line 290 - 293):
"The vegetation shows a high correlation for trees and desert with Ti/AL for the Sahel region (-0.734 and 0.781, respectively), and grass and desert for the Sahara region (-0.783 and 0.783). On the contrary, the correlations between vegetation coverage over the two regions is generally poor compared to the dust of Site 659, illustrating the strong non-linear relationship between vegetation and dust outside of the African continent."

A discussion of uncertainties in the forcings is missing. There are for example large uncertainties in the atmospheric CO2 concentration. Proxy reconstructions show a large uncertainty, particularly in the amplitude of 'glacial-interglacial' CO2 variability. The paper by Stap et al. 2016, just to name a model-based reconstruction where two of the authors of this paper are co-authors, shows a very different CO2 trajectory across the Pliocene-Pleistocene transition than that used in the simulations presented in this paper. I'm not saying that CLIMBER-2 should be re-run with all these alternative forcings, but a critical discussion of the possible impact that the choice of a particular forcing could have on the results presented in the paper is needed.

Yes, we agree. There are a couple of $CO_2$ reconstructions in the current literature that show a different behaviour. The same hold for proxy reconstructions, albeit these are not continuous records. The simulations with the Climber-2 run have been run prior to the work by Stap et al. (as the reviewer pointed out we were involved in both studies) and both come from different methodologies. We have added an additional paragraph in the discussion on this, also referring to a recent paper (still in discussion) which nicely shows a comparison of different $CO_2$ reconstructions (Figure 6 in Berends et al., CPD, 2020; doi: 10.5194/cp-2020-52). Added text (line 275 - 282):
" The climatic variability in the model is largely determined by changes in the forcing records; NH and Antarctic ice sheets, atmospheric greenhouse gas forcing of $CO_2$, and orbital variations (Laskar et al., 2004). Both timing and magnitude of the forcing will have an impact on the changes shown by the model. The ice-sheet forcing imposed here is based on a 3-D ice-sheet model constraint by the LR04 benthic d$^{18}$O stack by Lisiecki and Raymo (2005), which also determined the age scale of the $CO_2$ reconstructions. Ice-sheet changes can be different, but are constraint by the locations. On the other hand, $CO_2$ is much less constraint since proxy data over this time period are sparse, and model-based reconstructions can be quite different (see for example Figure 6 in Berends et al., 2020). Particularly, the reconstruction from Stap et al. (2016) shows a much larger amplitude in $CO_2$, whereas that from Willeit et al. (2019) employs a smaller amplitude over our time period."

**Minor comments**

lines 27-29: What is meant by 'completely'? There are plenty of other studies that could be cited here, showing that, at least if CO2 is low enough, orbital variations are enough to get pronounced glacial cycles: e.g. Abe-Ouchi et al., 2013 and Ganopolski & Calov 2011.

We meant that it can induce ice sheet growth, but not large ice sheet as seen during glacial maxima. This is the same as for example Ganopolski & Calov (2011) showed, for which large ice sheets do occur but only with low enough CO2 concentrations. We have revised the wording to (lines 28-30):
" Although radiative forcing of orbital variations is too small to force the world into or out of a glacial state with significant ice sheets, they are key to initiate ice-sheet growth and to pace glaciations (e.g. Bintanja and Van de Wal, 2008; Ganopolski and Calov, 2011)."

line 88: what does 'quality' mean here?

This mean how healthy the layer is, i.e. water content and thus prone to grow vegetation on it. We revised the sentence to (line 96-97):
"On the contrary, runoff and precipitation also depends on the water content and amount of vegetation that grows on the upper soil layer. "

lines 104-106: sentence is unclear

The sentence is revised to (line 116-117):
"When forcing records are kept constant we use the present-day ice sheet and a pre-industrial level of 280 ppm for $CO_2$."

lines 126-127: how has the tuning been done? Moreover, that the LR04 stack has almost no precession for the early Pleistocene.

As mentioned in Wang et al., 2010 (where the same data is used), the data for the last 2.6 Myr follow the same age scale, which is actually included in the LR04 stack (for benthic d18O). Between 5.2 and 2.6 Myr ago the data is retuned to minima in the Laskar et al. (2004) 65N insolation curve. This has been added to the text (line 138 - 140).

lines 136-139: Would be interesting to see the time series for precipitation, evaporation and runoff for the two grid cells. Also, what is happening to vegetation in these grid cells? Could it be that the increase in evaporation is related to an expansion of vegetation in the Sahara grid cell? If vegetation is growing over the Sahara I guess that more water should be available to evaporate because roots have access to deeper soil layers...?

We have added an additional figure (now Figure 3) that shows the vegetation fractions over time from 3.2 to 2.3 Myr ago. Figures of precipitation and evaporation are shown below. We have revised the text (line 155 - 162) as follows:
"For the Sahel region the runoff is strengthening following the African summer monsoon, driven by an increase in NH insolation. The increase in precipitation causes an increase in

trees, which replaces desert and grass (Figure 3b). This enhances evaporation, which is stronger than for grassland, causing the peaks in runoff. In contrast, the runoff values of grid box 12 (Sahara Desert) do not increase by the strengthened monsoon, but show peaks of low runoff during precession maxima. Although precipitation is enhanced during the summer monsoon when the air from the Atlantic Ocean reaches land, higher temperatures provide more room for water to evaporate, in combination with an increase of grass cover (Figure 3a) In the case of grid box 12, this additional precipitation is therefore compensated by an increase in evaporation. Also, during precession maxima precipitation is reduced and vegetation disappears, which leads to a strong decrease of evaporation and minima in the runoff."

[Figure]

lines 142-143: is this possibly related to changes in Atlantic meridional heat transport and subsequent changes in the position of the ITCZ when NH ice sheets start to grow and decay?

There is a link there, because increased precipitation is related to a northward shift of the ITCZ, but this holds for precession as well. With the data we have from the model we cannot fully investigate the link with heat transport and obliquity. The figure below shows the obliquity frequency (filtered at 0.0245 ± 0.003) of the four climatic forcing runs: orbit only (O: orange), orbit + CO2 (OG: green), orbit + ice sheets (OI: blue) and orbit + CO2 + ice sheets (OIG: red). The obliquity strength (i.e. amplitude) is mostly equal prior to 3.0 Myr ago for all 4 runs, when ice-sheet are included (OI: blue and OIG: red), there is a clear strengthening of the obliquity frequency in the Runoff. We have added a note on this in the paragraph that follows (line 176 - 177):
" Also, the power of the obliquity frequency of runoff is increased in the OI and OIG relative to the O and OG simulations."

[Figure]

Fig. 3: Please mention in the caption that the y-axis for precession is reversed in 3a. It took me a while to figure out that maxima where actually minima.

This is now Figure 4, this has been added, the same for other figures where it was missing.

lines 149-152: I have read this sentence 10 times, but still do not understand what it means.

We understand, the sentence is changed to (line 172 - 176):
"This shift can be attributed to the imposed lag in the tuning of the LR04 benthic d18O data when calibrating the depth-age scale. The time lag between obliquity  (41-kyr) and its related frequency component in the LR04 stack is gradually increased from 3 kyr prior to 3 Myr ago towards 5-6 kyr up to 1.2 Myr ago. This follows from an anticipated slower response time of the growth of larger Pleistocene ice sheets (Lisiecki and Raymo, 2005)."

line 155: 're-tuned age model of Sites 659'. It is not a spectrum of the age model, but of the dust record, right?

Yes correct, sentence is changed to (line 180):
".. and the dust record on the re-tuned age model of Site 659."

line 254: 'we correlation combined': rewrite
Changed to:
"Following, we correlated the combined runoff output of grid box 11 and 12 with the Ti/Al record."

line 257: 'which representing' -> representing
Agreed, removed 'which'

line 259: 'that indicating' -> indicating
Agreed, removed 'that'

Fig. 9d: and how are lags and leads represented? Please add a legend with arrow directions to clarify. Color scale is missing in d.

Yes, the colour scale is missing, but not needed for the interpretation, with orange-yellow within black lines as significant and strong power. Arrow direction is also explained in the caption. We have changed the caption accordingly.

**References**

Stap, L. B., de Boer, B., Ziegler, M., Bintanja, R., Lourens, L. J. and van de Wal, R. S. W.: CO2 over the past 5 million years: Continuous simulation and new d11B-based proxy data, Earth Planet. Sci. Lett., 439(April), 1–10, doi:10.1016/j.epsl.2016.01.022, 2016.

Bauer, E. and Ganopolski, A.: Aeolian dust modeling over the past four glacial cycles with CLIMBER-2, Glob. Planet. Change, 74(2), 49–60, doi:10.1016/j.gloplacha.2010.07.009, 2010.

Abe-Ouchi, A., Saito, F., Kawamura, K., Raymo, M. E., Okuno, J., Takahashi, K. and Blatter, H.: Insolation driven 100,000-year glacial cycles and hysteresis of ice-sheet volume., Nature, 500(7461), 190–3, doi:10.1038/nature12374, 2013.

Ganopolski, A. and Calov, R.: The role of orbital forcing, carbon dioxide and regolith in 100 kyr glacial cycles, Clim. Past, 7(4), 1415–1425, doi:10.5194/cp-7-1415-2011, 2011.

---

## Author Comment (AC2) · 16 Nov 2020

**Reply to the anonymous reviewer 2**

We would like to thank the reviewer for his/her comments. The remarks have definitely improved the manuscript. Below you will find a point-by-point reply, with the review given in **black** and our reply given in **blue**. We hope that we have answered all questions sufficiently. The line numbers we included in our answers refer to the marked changes document.

**General Comments**

The paper by de Boer and coauthors compares transient records of runoff from the Sahara and the Sahel that is produced by an intermediate complexity model with the dust record from 659 and the Ti/Al XRF record from 967 across the intensification of Northern Hemisphere glaciation. They find there these is strong precession-scale variability in all of the records, and that obliquity, thought to be driven by the high latitude ice sheets, has more of an impact on the Sahara following the intensification of NH glaciation. Authors also use a clever technique of correlating the empirical records with a range of combinations of grid cells to find the most likely combination of regional impact on the sediment and look at how this combination changes through time. The methodological steps are laid out in a fairly clear way with useful figures, but often the details of the model and time series bog down the interpretation. There is little conclusion on what the lags, relationship between runoff and dust, and the change in regional input to the Mediterranean really means in terms of climate dynamics, and also what it means for humans evolving in East Africa. Please find my specific comments with more detail, as well as some technical comments on the text and figures, below.

We thank the reviewer for his comments, please find below our reply to the specific comments.

**Specific Comments**

There needs to be more background in the first paragraph of the introduction. There are newer studies that have good enough age constraints to examine the phase relationships between orbital properties and the MPT and other transitions. I think authors also need to make it clear that the 100 kyr cycle after the MPT is likely an average of 120 and 80, as many have recently shown. Further, the MPT is not studied in this research, so perhaps removing that from the introduction would leave more room to provide better detail on other arguments.

Yes, we agree with the MPT characteristics. We will adjust the introduction accordingly, and mention the onset of NH glaciation instead at this stage of the introduction. This is also a key point of our paper, and will be more elaborately discussed in the introduction.

There also need to be far more citations for a sentence like line 39-41, where the authors link orbitally induced variability in the tropics. This has been seen in Africa in many different records by Tierney, Lupien (both Pliocene and Pleistocene), Rose, etc. In the conclusions (line 239-241) there is a vague sentence on this as well – orbital induced variations of what? If authors are talking about precession in climate records from Africa, there need to be many more citations.

Yes, we have extended this part of the introduction by showing that the changes are far broader. The new part is (lines 41-46):
"The orbital-induced variability is not limited to the high latitudes, it is also seen over the entire African continent. North African monsoonal records are linked with runoff and precipitation (e.g. Lourens et al., 2010), which persisted throughout the Pleistocene (Wagner et al., 2019). Changes are seen during the late Pliocene and mid-Pleistocene in vegetation in northeast Africa (Rose et al., 2016), during the Pleistocene in Kenya (Lupien et al., 2018), the West African monsoon (Kuechler et al., 2018) and hydroclimate variability in southeastern Africa over the past 2 million years (Caley et al., 2018)."

In the conclusions the sentence is revised to (lines 276-277):
"Previous studies have shown the strong presence of orbital induced variations of the African monsoon, that could .."

Similarly, the third paragraph of the intro on hominin evolution is very light in citations and seems to pick specific details from Joordens 2019 rather than focusing on the evidence and mechanisms for hominin transitions at the onset of NH glaciation or the variability selection hypothesis (Potts).

Yes, this bit is indeed weighting on the paper of Joordens et al, since she was involved in the earlier part of this project, as an expert on this specific topic. Therefore, also the reasoning for this paragraph. We have tried to extend it a bit more with adding more references, although the mentioned is rather general, since not our expertise (lines 49-57) :
"Specific climate transitions, among which the Plio-Pleistocene transition, coincide with the possible emergence or extinction of hominin species (Donges et al., 2011). ...
Condition of highly variable climate and strong seasonality during eccentricity maxima would result in isolated refugium for early hominins, that would be conclusive for evolution (Trauth et al., 2007). ...
However, the evolution of the hominin species throughout the Pleistocene is a highly complex process (Mounier and Mirazón Lahr, 2019)."

'Continental runoff (i.e. p-e)' is a bit misleading. This p-e balance is not equivalent to runoff as water can be stored in soil, lakes, and groundwater, which is fully stated in the description of the model. This is brought up again in line 129 and onward and needs to be explained more clearly and thoroughly before the interpretations are stated.

Yes, we understand that this relationship is not as straightforward as we imply here. We will explain this in better detail in the introduction, changes are made in lines 58-59:

" In this paper we focus on the connection between the African monsoon, using continental runoff (linked to precipitation, evaporation, and water storage in the soil, lakes and groundwater)."

And lines 142-144:
" Runoff over the Northern African continent as modelled in CLIMBER-2 largely results from the difference between precipitation and evaporation over land, although water can also be stored in the soil, lakes and groundwater."

Figure 1 has the model grid cells, but they are not plotted over a map based on the 6 potential surfaces. This would be helpful.

Yes, we understand that could be helpful, but we would like to focus the map on Africa, since potential surfaces also include ice sheets and sea ice, which can only be viewed on a global map.

Why not use the principal component analysis, based on Ti/Al, from Grant 2017 for the 967 comparison? This is brought up later, and perhaps is a better estimate of different line types for the different frequencies, and in c, use blue.

The main reason behind this is that we are presenting an extension of the Ti/Al record of Site 967. The Grant (2017) wet-dry index also does not extend before 3.0 Myr ago. Secondly, the wet-dry index is a product of XRF scanning and a PC analysis, so less straightforward as the terrigeneous Ti/Al data.

Line 131: the location of the Sahel today is shown in Figure 1, but the location, spread, vegetation may well have been very different in the past, particularly in the Pliocene. How does this affect the grid cell coverage?

Following from a comment of Reviewer 1, we have added a figure that shows the distribution of the vegetation fractions over time from 3.2 to 2.3 Myr ago. This figure clearly shows that gridbox 11 shows a vegetation distribution consistent with the Sahel. When introducing the figure we will mention this, referring to Figure 1 (lines 149-151):
" Both gridboxes show high variability of the vegetation fraction, but do show a clear linkage to the present-day coverage of partially desert and vegetation for gridbox 11 and desert for gridbox 12 (Figure 1)."

The discussion surrounding the lags seems a bit tangential and perhaps unnecessary. If the lag is due to a choice in the LR04 tuning, then does this analysis tell us anything new?

Yes, a large part of the records being in phase is due to the tuning (relative to LR04). We do think it is important to inform the reviewer that when including the ice-sheet forcing an additional lag is introduced in the system. This lag is than even seen in low latitude records, we do think that that is worth mentioning.

Line 207: the power spectra should not be influenced by lack of data. Either resample the data appropriately, or put the constraints up front in the methods section. Can authors resolve precession throughout the records? If not, then conclusions based on evolutive spectra shouldn't be made, or should be modified.

Yes, we agree. this sentence has been changed to (lines 241-245):
"For the dust record of Site 659, besides the eccentricity minima a dip in the 23-kyr power around ~2.6 Myr ago is seen (Figure 7b,d), for which we would expect 23-kyr power to stay high following the 400-kyr eccentricity maximum (Figure 2a). This also underlines the non-linear link between dust and runoff as illustrated in Figure 8."

What are the conclusions of the mechanisms linking the ice sheet inception to the change in runoff region source? The conclusions section appears to be more of a summary. Try to utilize this space for connections back to the topics brought up in the introduction – what about human evolution? Would this shift, and other aspects of the findings, impact humans, and how?

Following a comment of reviewer 1 we have added a full paragraph in the Discussion on the forcing records, naturally influencing the outcome of the CLIMBER-2 model simulations. We have added additional sentences on the link to the shift after NH ice sheets increase in size, and the link to human evolution (lines 291-292):
"Although the evolution and dispersal of hominin species during this time could be linked to orbital variations (Joordens et al., 2019), a direct link with the climatic variations shown by CLIMBER-2 cannot directly be established."

**Technical Corrections**

"Myr age" can be replaced by "Ma" in every circumstance
We would rather stick to our common usage of Myr ago.

Line 42: replace 'deterioration'
Replaced with 'variability'

Line 42: 'central' Africa? Do authors mean East Africa?
Yes, corrected

Figure 3: the colors need to match better for clarity. Perhaps in b, use orange with different line types for the different frequencies, and in c, use blue.
We would like to stick to the colours as is, since we use the same colours in Figure 7 (which in the new version is the new figure 9).

If Table 1 could be shown in a figure, I think the point of lags would be much easier to comprehend by the reader. Perhaps use phase wheels.
Since it is only a minor part of the discussion, we would keep Table 1 as is.

Figure 6: the legend should be changed to either show the location or the proxy, not the proxy for one site, and the site number for the other.
Changed

Try not to conflate the Plio-Pleistocene transition with the onset of NH glaciation – authors have cited these at two distinct times (2.6 vs 2.8 Ma), so try to keep the wording consistent.
Yes, we have checked this through the text

There are multiple instances of awkward phrasing, run-on sentences, and misused gerunds, so further editing for grammar could benefit the clarity of the manuscript.
Yes, we will go through the new version of the manuscript thoroughly.